# Discovering High-Quality Chess Puzzles Through One Billion Plays with Offline Reinforcement Learning

## Abstract

Learning and skill mastery requires extensive and deliberate practice. In many learning settings, producing high-quality pedagogical materials can require a high level of domain expertise and be very time-consuming. Pedagogical materials often need to train students to engage in different thinking patterns. In some domains, such as chess, puzzles are used to help students practice their skills in calculating the next moves and recognizing known patterns on a board. Giving students a practice set of puzzles to help them learn different modes of thinking is challenging because the teacher needs to carefully balance between different motifs and how many look-ahead steps a student needs to perform. Popular online platforms like Chess.com and Lichess offer players millions of puzzles. Unlike chess tactics puzzles procured by human experts, where chess beginners can learn valuable insights, these puzzles are automatically generated and often regarded as having low pedagogical values. These platforms also rely on a heuristic to recommend puzzles to users for practice. Using the user history data over an entire year, a total of 1.6 billion puzzle-solving histories, we learn the pedagogical value of a puzzle and how to automatically choose a set of puzzles to better support chess learners in a completely unstructured way using insights from offline reinforcement learning. We validate the quality of the puzzles discovered by our model by collecting annotation ratings from titled chess players. The success of our pipeline shows promise for a future where we can understand the pedagogical values of practice items in other domains like math or coding problems.

## 1 Introduction

Practice makes perfect. The foundation of gaining knowledge or mastering a new skill relies on countless hours of mindful and delibrate practice (Ericsson et al., 1993; Anders Ericsson, 2008). Koedinger et al. (2023) suggests that across many different laerning settings, when students are provided with high-quality, curated learning materials, they can all learn at a similar rate and achieve success with extensive practice. However, the creation of high-quality learning materials is often a key bottleneck. Though lectures can be recorded in video and knowledge can be transcribed in text, students need to be able to practice what they have learned through forced retrieval and synthesis, which has been shown to greatly enhance learning (Roediger III et al., 2011). Producing a large amount of practice materials often requires high human expertise and heavy time investment. It is also unclear how effective some of the learning materials are in terms of helping students achieve mastery of certain skills.

In chess, players often learn by playing against each other directly. However, they also learn important skills through tactics books, which are comprised of puzzles – subgames limited to a few moves to teach important concepts (called motifs) or to train players to plan multiple moves in a sequence in order to gain more advantage over their opponent. These puzzles are very common for beginners and are considered good learning materials because they isolate and highlight difficult concepts into a small subgame and prime beginners into a habit of thinking strategically (Henkin, 2021). Online chess platforms such as Chess.com and Lichess provide puzzles for players to practice. In order to keep their players engaged, these online platforms aim to serve fresh puzzles on a weekly basis so that players will always have new materials for learning and fun. Chess.com serves roughly 0.5M puzzles to their players and Lichess hosts over 3.8M puzzles.

In order to produce a large number of new puzzles quickly, unlike classic tactics books where puzzles are curated by human experts through careful deliberation, review, and editing, online chess platforms use an algorithm to automatically generate puzzles from the actual games played by players during that week. On a high level, this automatic algorithm captures some human intuition that each puzzle needs to have a sequence of best moves, where deviating from the best move will result in a large decrease in the player's win rate in that game. The puzzles are assigned with an Elo rating and then served to players randomly if the difficulty is somewhat within the range of the player. We do not know whether these puzzles are high-quality teaching materials – whether they are effective at increasing the long-term chess knowledge of the player. In a system where learning materials can be automatically generated, it is crucial to filter out materials that do not lead to an effective long-term increase in knowledge or skill.

Assessing the quality of a learning material requires two steps. First, we model the dynamics of a player's knowledge growth: a puzzle that increases the user's internal knowledge is more valuable than a puzzle that does not. We then need to know which puzzles are most effective at increasing a player's knowledge gain, where a player can gain knowledge faster by solving the right sequence of puzzles. Offline reinforcement learning (RL) (Levine et al., 2020) has proposed a suite of tools to learn and evaluate a policy through past data. Recommending chess puzzles to players can be modeled as a policy that decides on which puzzle to serve. Combined with a deep knowledge tracing model, we show that we can discover high-quality chess puzzles using historical data alone.

In the era of generative models, where large foundation models can automatically produce a massive amount of educational materials at a moment's notice, it becomes increasingly crucial to understand which materials are more valuable and effective for learning than others. Using a large dataset from Chess.com, which contains 2.9M users and their 1.6 Billion puzzle interactions over a year, we are able to propose a learning material assessment model that combines deep knowledge tracing and offline reinforcement learning. We designed a chess puzzle quality annotation schema and recruited United States Chess Federation (USCF) titled players to validate our model. We show that our model can serve valuable and effective puzzles for players with some history of interactions and a known rating, a valuable contribution to the chess community. Even though we only demonstrated success in a single domain of chess puzzles, we expect the possibility of building such a system in other domains of learning, such as language, math, and coding.

## 2 DATA

Our dataset consists of players' puzzle history data from 3,132,428 unique, active users of a popular chess website, Chess.com, playing a total of 1,536,254,297 puzzles (441,113 unique puzzles) over the course of one year from March 2021 to March 2022. On average, a user played 490.4 puzzles over this one year. Of those 490.4 puzzles, 96.9% are played within three minutes of another puzzle, which indicates that players tend to play puzzles in short bursts of time. An average burst comprises 5.1 games. Playing in bursts naturally suggests that users do not continuously play puzzles throughout the day. In fact, the average time between these bursts is approximately two and a half days. In general, our dataset reflects the engaged and extensive user base of online chess players.

**Puzzle Serving**   According to Chess.com administrators, the site serves puzzles to users using a bucketed uniform policy. This policy serves a player's next puzzle by first bucketing all chess puzzles that have puzzle ratings within $\pm 200$ points of a player's ELO and then sampling uniformly from that bucket. As a player begins to fail puzzles, that $\pm 200$ bucket eases to $-300/+100$ after one incorrect puzzle, $-400/+0$ after two, $-500/-100$ after three, and $-600/-200$ after four incorrect puzzles. In this way, the Chess.com policy adapts to player performance by serving progressively easier puzzles in response to a player's difficulty in solving previously served puzzles. Overall, Chess.com's *bucketed uniform policy* integrates a player's performance on their past four puzzles, their ELO rating, and puzzle rating to determine the next puzzle served.

**Chess Puzzle**   Chess puzzles are core units of learning. In a chess puzzle, the player is presented with an initial position and tasked with finding the correct solution in one or more moves, adhering to the puzzle's main goal or task, described in part through a puzzle's motifs. The intial position is shown to the player as a board, and it can also be conveniently encoded as a FEN (Forsyth-Edwards Notation) string. Move counts for puzzles in our dataset range from 1 to 16, averaging at 2.6. Harder puzzles tend to demand more moves, with the puzzle's difficulty reflected in its puzzle rating. Our

puzzles' ratings range from 100 to 4000. In contrast to move count and puzzle rating, which are both numerical, motifs provide more qualitative insights into the diversity of puzzles.

A puzzle's motifs describe the primary strategy or skill required to solve the puzzle. Examples of such motifs include "Exchange Sacrifice" and "Promotion," describing a notable event that takes place in a puzzle. A puzzle can have more than one motif; in fact, the puzzles in our dataset have, on average, 5.6 motifs each, with motif count ranging from no motifs at all to 56 motifs. Of our 589,720 puzzles, 26% do not have any motifs at all. Thus, motifs provide informative yet nonexhaustive insights into the quality and variety of served puzzles.

**Analyzing User ELO Growth** In Figure 1a, we plot the mean change in rating across the first 50 recorded plays for four groups of players based on their ELO ranges and relative increases in ELOs. The "Growth Group" represents players in the 99th percentile for ELO increase (the change in ELO from each player's first recorded ELO) among players in their ELO range, while the "Stagnant Group" represents players in the 1st percentile for ELO increase, combined across all three ELO ranges. The trajectory of the Stagnant Group, as expected, is relatively flat, signifying little growth in ELO across the first 50 games played. On the other hand, players in the Growth Group of high ELOs improve rapidly in their first 50 games, with less rapid growth for lower ELO players, and even less rapid growth for the lowest ELO players. This suggests that a player's average, long-term ELO generally points to their initial improvement speed. Interestingly, the Growth Group of the lowest ELO players experiences a dip in ELO in their first ∼40 games, falling below ELO improvements of even the Stagnant Group, before seeing gains. This dipping trend suggests that, for overall inexperienced players, playing chess puzzles involves an especially difficult acclimatization period, during which unfamiliarity with the game may initially hinder performance, but players eventually learn the rules and complexities of the game and reap the benefits of playing more puzzles.

We broaden our analysis in Figure 1b and examine players' total puzzle counts. Each point in the figure represents an individual player's total number of puzzles played. We observe a general trend from the bottom left toward the upper right, indicating that a higher number of puzzles played generally correlates with a higher ELO. This strengthens the suggestions that playing chess puzzles is a skill that improves with more experience. The total number of puzzles played is generally an indicator of a player's skill level.

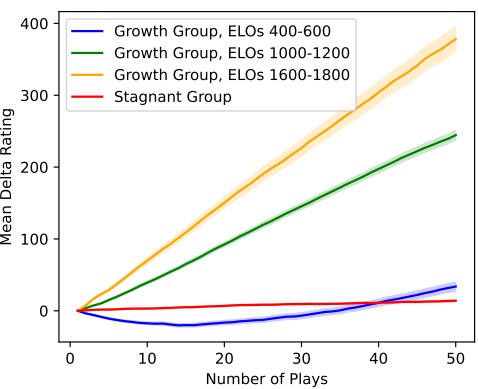 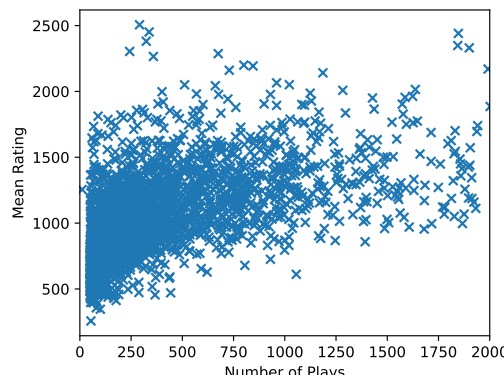

(a) Plot illustrating the mean change in rating for four groups of players based on their ELO ranges and their relative increases in ELOs.

(b) Scatter plot showing the relationship between a player's puzzle play count and their rating averaged over a year.

Figure 1: Plots that illustrate user ELO growth, showcasing the ELO growth trajectories of several groups of players (Figure 1a) alongside overall ELO trends across our entire player base (Figure 1b).

## 3 RELATED WORK

There is a long history of using games such as Chess to evaluate the progress of AI (Campbell et al., 2002; Silver et al., 2017; 2018). Although these models have developed superhuman abilities to master the game, few investigations have focused on leveraging their knowledge to teach humans. Schut et al. (2023) and McGrath et al. (2022) did pioneering work on uncovering chess knowledge

learned in AlphaGo and AlphaZero. McIlroy-Young et al. (2021) built models to identify the chess playing styles of each user and then later released models that imitate chess players of different levels (McIlroy-Young et al., 2022). But largely, there have not been end-to-end systems that can successfully transfer knowledge learned in models to teach humans.

On the building of automated systems to teach front, reinforcement learning has been used to create systems that can adaptively select teaching materials for students. Ruan et al. (2024) built a math tutoring tool that decides when to provide hints. Mandel et al. (2014) uses offline RL to augment the fraction learning experience of students in math games. However, most of these projects remain small in scale, and offline RL has not demonstrated usefulness for large-scale educational settings. In our work, we follow the lead of Kumar et al. (2022) to scale offline RL models to large datasets with billions of interactions and use offline RL to discover high-quality puzzles that can improve a chess player's learning experience.

## 4 POLICY LEARNING

### 4.1 PRELIMINARIES

We define a stochastic decision process $M = \langle \mathcal{S}, A, T, r, \gamma \rangle$, where $\mathcal{S}$ is a set of states; $A$ is a set of discrete actions; $T$ is the transition dynamics; $r$ is the reward function; and $\gamma \in (0, 1)$ is the discount factor. Let $D_n = \{\tau_i\}_{i=1}^n = \{s_i, a_i, s_i', r_i\}_{i=0}^n$ be the trajectories sampled from $\pi$ on $M$. We denote the performance of a policy $\pi$ based on its expected discounted return $R_t = \mathbb{E}_{\tau \sim \rho_\pi}[\sum_{t'=t}^T \gamma^{t'} r_{t'}]$ where $T$ is the time horizon and $\rho_\pi$ is the distribution of $\tau$ under policy $\pi$. By the definition of the action-value and value function repsecively, we let $Q(s, a) = \mathbb{E}_{\tau \sim \rho_\pi}[R_t \mid s, a]$ and $V(s) = \mathbb{E}_{\tau \sim \rho_\pi}[R_t \mid s]$.

In an off-policy policy learning problem, we do not have access to the true transition dynamics $T$ or any online interaction with $M$. Instead, we take an offline dataset $\mathcal{D}$, which can be collected by one or a group of distinct policies, which we collectively refer to as the behavior policy $\pi_\beta$ on the decision process $M$.

In actor-critic frameworks, we perform policy evaluation and policy improvement in conjunction to derive an optimal policy $\pi^*$ (Konda & Tsitsiklis, 1999). Typically, policy evaluation is performed using iterative application of the Bellman update. In the context of deep RL for off-policy learning (i.e., where we have $\mathcal{D}$, policy $\pi_\theta$ parameterized by $\theta$, action-value function parameterized by $\phi$), we perform policy improvement to gradient updates to the policy $\pi_\theta$ and optimize Equation 4.1.

$$\arg\max_\theta \mathbb{E}_{\mathbf{s} \sim \mathcal{D}, \mathbf{a} \sim \pi_\theta(\cdot|\mathbf{s})}[Q_\phi^\pi(\mathbf{s}, \mathbf{a})]$$

For our formulation, we consider a non-Markovian policy $\pi_\theta$ based on a transformer architecture, where the state and action space incorporate a fixed length context. Specifically, our state space consists of a continuous embedding-based $k$-dimensional representation of the user's puzzle history and learning progress (e.g., ELO), i.e., $s \in \mathcal{S} = \mathbb{R}^k$. Our desired action space is the set of all $N = 589,721$ chess puzzles, i.e., $a \in \mathcal{A} = \mathbb{N}^N$.

In an off-policy setting, common pitfalls with traditional actor-critic techniques include extrapolation error by venturing outside of the supported data distribution in $D_n$, which results as an accumulation of errors from bootstrapped action-value functions (Sutton, 1999). To ensure that the trained policy $\pi_\theta$ stays close to the prior policy, we penalize the Kullback-Leibner divergence between the trained and behavioural policy with factor $\beta$ (Rafailov et al., 2024; Tutor).

$$\arg\max_\theta \mathbb{E}_{\mathbf{s} \sim \mathcal{D}, \mathbf{a} \sim \pi_\theta(\cdot|\mathbf{s})}[Q_\phi^\pi(\mathbf{s}, \mathbf{a})] + \beta D_{\text{KL}}(\pi_\theta \,||\, \pi_\beta)$$

### 4.2 DERIVING OFFLINE POLICY LEARNING OBJECTIVE

We derive a simple advantage-weighted actor-critic-style objective for offline policy learning, based primarily on Nair et al. (2020) and Kostrikov et al. (2021). We leverage offline advantage estimation via a parameterized value function $V_\psi^\pi(\mathbf{s})$, augmenting the objective shown in Equation 4.1 with a baseline. Importantly, note that since the value function and its inputs are constants with respect to the optimization variable, $\theta$, it does not bias or modify the objective.

$$\arg\max_\theta \mathbb{E}_{\mathbf{s} \sim \mathcal{D}, \mathbf{a} \sim \pi_\theta(\cdot|\mathbf{s})}[Q_\phi^\pi(\mathbf{s}, \mathbf{a}) - V_\psi^\pi(\mathbf{s})] + \beta D_{\text{KL}}(\pi_\theta \,||\, \pi_\beta)$$

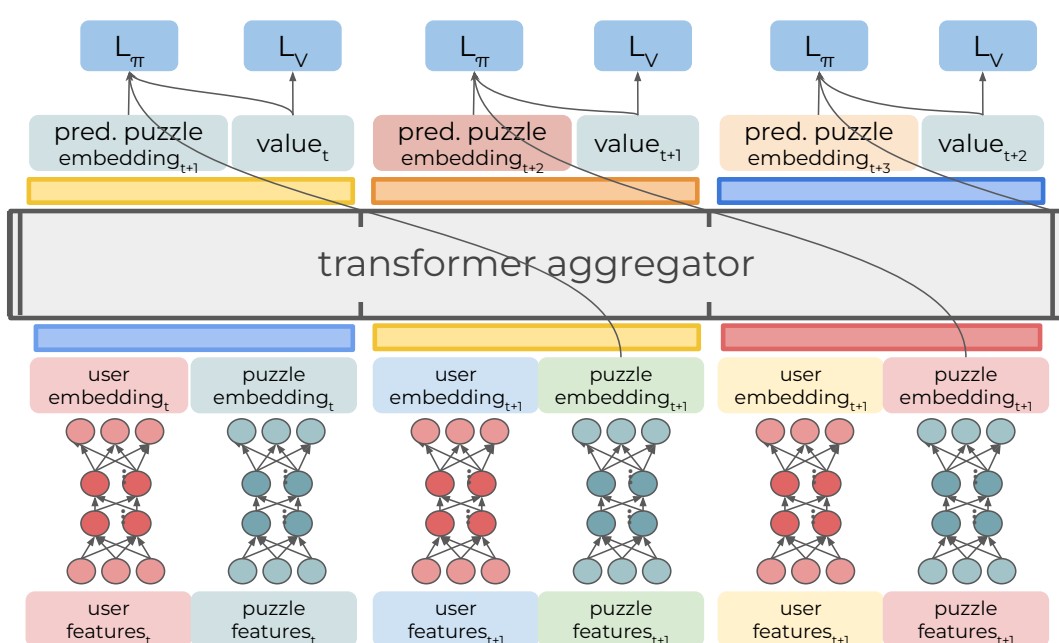

Figure 2: Architecture and training procedure of the chess puzzle recommendation policy, leveraging user features and puzzle features to optimize a policy-based loss $L_\pi$ and a value function via $L_V$.

Based on the derivation in Nair et al. (2020), we can project the closed-form optimal solution $\pi^*(\mathbf{a}|\mathbf{s}) \propto \pi_\beta(\mathbf{a}|\mathbf{s}) \exp(\frac{1}{\beta}(Q_\phi^\pi(\mathbf{s}, \mathbf{a}) - V_\psi^\pi(\mathbf{s})))$ into the policy space through minimizing the KL-divergence, which simplifies into a simple weighted maximum likelihood objective, shown in Equation 4.2.

$$L_\pi(\theta) = \mathbb{E}_\mathcal{D}[-\log \pi_\theta(\mathbf{a}|\mathbf{s}) \exp(\frac{1}{\beta}(Q_\phi^\pi(\mathbf{s}, \mathbf{a}) - V_\psi^\pi(\mathbf{s})))]$$

Mirroring Kostrikov et al. (2021), we train the value network and action-value network for advantage estimation purely using transitions in the dataset $\mathcal{D}$. To train the value network, we leverage expectile regression, as shown in Equation 4.2, and to train the action-value network, we use the objective shown in Equation 4.2.

$$L_V(\psi) = \mathbb{E}_\mathcal{D}[L_2^\tau(Q_\phi^\pi(\mathbf{s}, \mathbf{a}) - V_\psi^\pi(\mathbf{s}))]$$

$$L_Q(\phi) = \mathbb{E}_\mathcal{D}[(r(\mathbf{s}, \mathbf{a}) + V_\psi^\pi(\mathbf{s}') - Q_\phi^\pi(\mathbf{s}, \mathbf{a}))^2]$$

### 4.3 ARCHITECTURE FOR CHESS PUZZLE RECOMMENDATION

Since we formulate the recommendation problem as non-Markovian offline policy learning, we use a transformer decoder architecture (i.e., casual transformer) to power our decision making policy, denoted by $T_{\theta_T}$. At each sequence element of the transformer, we input a concatenation of a user representation $u_t$ and puzzle representation $p_t$ at the current time, constructed through applying a neural network on user and puzzle features respectively. The corresponding outputs at each sequence element are (a) the prediction of the next puzzle $\pi(\mathbf{a} \mid \mathbf{s})$ and (b) the value function estimate $V(s)$.

During training, we optimize $L_\pi$, $L_Q$ $L_V$ on predictions at each sequence element, with training sequences sampled uniformly from the chess puzzle interaction data. Importantly, we apply teacher forcing during the forward pass of training, which is standard in training autoregressive decoder-only transformers.

**User Embedder** To construct a latent representation of the user, we embed user-specific features $f_u$, i.e., the user ELO and user correctness at time $t$, using a small, two layer multi-layer perceptron (MLP), denoted as $U_{\theta_u}$. We normalize the output user embedding $u_t = U_{\theta_u}(f_u)$ to unit norm (e.g., L-2 normalization).

**Puzzle Embedder**   To construct a latent representation of the puzzle, we embed puzzle-specific features $f_p$ using an MLP and convolutional neural network (CNN), denoted as $P_{\theta_p}$. Specifically, we use a CNN to encode the board position associated with the puzzle, alongside a learned embedding for each individual puzzle and its first move. Similarly to the user embeddings, we normalize the puzzle embedding $p_t = P_{\theta_p}(f_p)$ to unit norm (e.g., L-2 normalization).

**Practical Considerations for Policy Learning**   Given a large discrete action space $\mathcal{A}$ (i.e., approximately half a million puzzles), the policy probabilities $\pi_\theta(a \mid s)$ are challenging to compute and leverage to compute $L_\pi$ tractably (i.e., it requires computation and normalization across the entire action space). Instead, we leverage the pre-computed puzzle embeddings for the next timestep (e.g., at $t + 1$ given prediction at timestep $t$) for our prediction task by reframing the policy as an exponentiated inner product of the transformer's output and the puzzle embedding. As shown in Equation 4.3, the policy probabilities simply reduce to a temperature-weighted softmax applied to the dot product between the predicted next puzzle embedding (from $T_\phi$) and the true next puzzle embedding $p_t$.

$$\pi_\theta(\mathbf{a_t} \mid \mathbf{s_t}) \propto \exp(\lambda \cdot T_{\theta_T}(\mathbf{p}_{t-T..t}, \mathbf{u}_{t-T..t})^\top p_{t+1})$$

Despite the above simplification, during training time, it is still intractable to estimate the probabilities of the observed action $a_t$ from $\mathcal{D}$ directly since it would require a normalization step across all action (puzzle) embeddings (i.e., forward pass on all puzzles' features to compute their puzzle embeddings). Consequently, we make a further simplification to estimate the probability through an in-batch "negative" sample of puzzles, where we compute the normalizing factor for Equation through random sampling. Given a batch $B$ containing a set of puzzle embeddings $P$, we estimate the policy probabilities as shown in Equation 4.3.

$$\pi_\theta(\mathbf{a_t} \mid \mathbf{s_t}) \approx \frac{\exp(\lambda \cdot T_\phi(\mathbf{p}_{t-T..t}, \mathbf{u}_{t-T..t})^\top p_{t+1})}{\sum_{p_n} \exp(\lambda \cdot T_\phi(\mathbf{p}_{t-T..t}, \mathbf{u}_{t-T..t})^\top p_n)}$$

Afterwards, we apply the losses $L_\pi$, $L_Q$ and $L_V$ to optimize the policy with respect to the transformer parameters and user and puzzle embedders. We summarize the entirety of the training algorithm in Algorithm 1 and as depicted in Figure 2.

---

**Algorithm 1** Training algorithm for off-policy learning on chess puzzles.

---

**Input:** Dataset $\mathcal{D}$, causal transformer $T_{\theta_T}$, user encoder $U_{\theta_u}$, puzzle embedder $P_{\theta_p}$.
**for** each batch of user sequences $B \subset \mathcal{D}$ **do**
    extract puzzle-only features $f_p$ and user-only features $f_u$ from batch $B$
    compute puzzle embeddings $p_t = P_{\theta_p}(f_p)$ and user embeddings $u_t = U_{\theta_u}(f_u)$
    compute approximate probability of observed next puzzle $\pi_\theta(\mathbf{a_t} \mid \mathbf{s_t})$ using in-batch sampling
    compute value $V_\psi^\pi(\mathbf{s}')$ and action-value function $Q_\phi^\pi(\mathbf{s}, \mathbf{a})$
    compute and backpropagate loss functions $L_V$, $L_Q$, and $L_\pi$
**end for**

---

## 4.4 Deriving Reward Function

During evaluation and practical usage, we compute puzzle embeddings across all puzzles to compute the probability distribution across the entire action space. While this requires computation of all puzzle embeddings, these can be reused independently for any predicted puzzle embeddings (i.e., it is upfront cost, but the embeddings need not be reinferred for each user sequence).

We construct a scalar reward function that attempts to quantify the learning benefit in terms of the user's correctness on the puzzle and relative puzzle difficulty. Trivially, a correct response on a challenging puzzle is ideal, and it highlights progression in terms of learning, warranting a large reward. In any cases where the user answers incorrectly, it demonstrates no verifiable evidence of learning, so we assign no reward in these cases. Otherwise, in cases of partial correctness, we assign rewards based on the proportion of correct moves and relative difficulty of the puzzle.

Given the number of total moves $N$, number of correct moves (from the user) $c$, and the ELO of the puzzle and user, we assign the reward based on the difficulty-weighted correctness, with the difficulty weight $\alpha$ being a hyperparameter.

$$r(s, a) = \frac{c}{N} \exp(\alpha \cdot (\text{ELO}_{\text{puzzle}} - \text{ELO}_{\text{user}}))$$

We provide a visualization of the reward function across different proportions of correct moves $c/N$ in Figure 3 for $\alpha = 0.002$, which is the chosen hyperparameter due to its reasonable balance between partial correctness and difficulty. Specifically, based on internal testing, $\alpha = 0.002$ provides a reasonable level of reward equivalence between partial correctness levels (e.g., halfway correct) on challenging puzzles and complete correctness on easier puzzles (e.g., with 200-400 less puzzle ELO). These equivalencies are displayed via dotted lines, where the color corresponds to the maximal reward achieved at a particular correctness level, and the intersection with other lines indicates the ELO delta at the reward equivalence.

Figure 3: Reward function $r(s, a)$ for varying levels of correctness and ELO differentials between puzzle and user, with reward equivalences shown between balances of correctness and relative difficulty.

## 5 SCALING EXPERT EVALUATION

While it would be ideal to run an experiment with the proposed new way of selecting puzzles and evaluate their effectiveness on chess learning, there are many challenges to this. It is expensive and hard to recruit the sample size needed, and it may take a significant time to be able to measure any difference reliably. Instead, as is often done in machine learning, we use expert annotators to bootstrap a system that can scale our annotation effort to a larger size.

### 5.1 EXPERT ANNOTATION

To annotate the quality of chess puzzles, we designed a detailed rubric with two chess experts involved in the project (USCF rating around 1900 and 2400). The rubric covers different characteristics of a puzzle, from whether it tests the player's ability to perform in-depth calculations of sequential moves, or it focuses on testing the ability to recognize iconic patterns. In addition, the experts are also asked to judge if the puzzle is appropriate for a player of a given rating, whether it is a high-quality puzzle, and if it is fun to play. The rubric took multiple iterations with internal testing to verify its robustness and is provided as a PDF document in the supplement.

| Category | Range |
|---|---|
| Calculation | 1-4 |
| Pattern | 1-3 |
| Informativeness | 1-5 |
| Appropriateness | 1-5 |
| Quality | 1-5 |
| Fun | 1-5 |

Figure 4: Expert rating categories. Ratings were later converted to 100 for LLM.

We recruited 8 chess experts to annotate 30 puzzles. The annotators are composed of 8 strong chess players, 7 of whom hold official chess titles. In total, there are 2 USCF (United States Chess Federation) Experts, 1 USCF National Master, 1 FIDE (Fédération Internationale des Échecs / The International Chess Federation) Master, 1 FIDE International Master, and 2 FIDE Grandmasters. The chess annotator's USCF ELO rating is between 1990 and 2576. For context, the chess legend Magnus Carlsen has an FIDE rating of 2832. The 100th highest-rated player has a FIDE rating of 2637 as of Fall 2024.

The puzzles are randomly sampled from a *bucketed uniform policy* based on the player's rating range, and we have 6 players with different Elo scores. We chose 12 puzzles as *preference learning* puzzles that are annotated by all experts. This means that out of the 30 puzzles each expert annotated, only 18 puzzles are unique between them. The annotation is performed through an anonymized spreadsheet. Each annotator is asked to spend 2 hours on the annotation and is later compensated with a gift card.

### 5.2 SCALING ANNOTATIONS WITH LLM

Judging whether a chess puzzle has high quality or not requires annotators with significant expertise, who are distinctly different from standard crowd workers. This also means we are not able to perform massive annotations that can take up to hundreds of hours. However, sometimes, two policies that learn to recommend puzzles can behave very similarly and, therefore, require a lot of annotations to understand if they are statistically significantly different. To help with scaling up the annotation effort, inspired by Zheng et al. (2023); He et al. (2023), we use the expert annotations to calibrate an LLM that can imitate expert chess annotator's judgments. Anecdotally, LLMs seem

to be able to understand FEN strings and can play chess to some extent[1][2]. These evidence gave us belief that LLM has some preliminary understanding of chess.

With some example expert annotations, we choose to use DSPy (Khattab et al., 2023) to create our LLM chess puzzle judge. DSPy is an LLM library that allows us to specify a set of initial prompts, training data, and a metric. It then automatically selects the best expert examples to include in the LLM prompt that can maximize the metric. Internally, DSPy performs cross-validation with a random search to find the best subset of expert annotations.

**Training Data** We learn 8 LLM annotators using the data from 8 chess annotators. We separately learn 8 annotators is because each annotator's preference might be different and imitating the behavior of 8 individual seems more difficult than imitating the behavior of one individual with their own data. Since each annotator produced 30 puzzle annotations, we split this into a training set of 12 puzzles, a validation set of 9 puzzles, and a test set of 9 puzzles.

**Prompt Design** We use the DSPy Signature module to design the prompt. We specify three fields: annotation guideline, the persona of the annotator (USCF ELO rating and title), user profile, and puzzle board. We provide a complete description of each field in the appendix. For puzzle board representation, we experimented with FEN string and a grid-like representation of the entire board (with piece names and positions). We found the grid-like representation to work better. Along with the board position, we also describe what the first move of the puzzle is in text.

**Optimization Metric** Our task is a rating task, where experts rate each puzzle for categories in Table 4. Therefore, the metric we define is the average over mean absolute error (MAE) of each rating category. DSPy can only maximize a metric; therefore, we maximize the negative MAE.

## 6 EXPERIMENT

To train our chess puzzle recommendation policy, the dataset is partitioned randomly by users, with 90% used for training and 10% for evaluation. During training, each batch consists of 256 examples of users' puzzle histories, with a sampled sequence length of 256 to accommodate computational constraints. Consequently, we use roughly 65K puzzles to estimate our policy's probabilities for training (approximately 10% of all puzzles). We train the policy for 25K, alongside the learned action-value and value functions.

**Evaluation using Importance Sampling** To evaluate our trained policy with respect to the behavioural policy, we can leverage importance sampling, using the known probability distributions of the behavioural policy (e.g., as deployed by chess.com) and our trained policy. Importantly, traditional importance sampling with sequences of size 256 (or even a fraction of that) is impractical due to the significant induced variance in the estimator over large sequences. Instead, we employ a one-step approximation (e.g., removing past weights) across 32-steps, which significantly reduces variance at the cost of being biased (Chen et al., 2019). To further reduce variance, we clip the importance weight for each step $\pi_\theta(\mathbf{a}|\mathbf{s})/\pi_\beta(\mathbf{a}|\mathbf{s})$ between $\epsilon_s = 0.1$ and $\frac{1}{\epsilon_s} = 10.0$.

To quantify our improvements over the behavioural policy for different users, we break down our results into different ELO groups. Additionally, we examine groups of stagnant growth and different growth groups, similarly to Figure 1; for this analysis, we use a simple definition of stagnant growth, which is growth of no more than 50 ELO points across the sequence. Importantly, this allows for examination of our improvements on different types of users and learning patterns, e.g., improvements on stagnant or new users may be more critical to retention on the chess.com platform.

## 7 RESULTS

In this section, we examine the performance of our learned policy in two different ways. We use an offline policy evaluation method to provide an objective comparison between our policy's performance and the behavior policy's performance. Then, we use our automated annotation pipeline to provide another set of ratings.

---

[1]https://nicholas.carlini.com/writing/2023/chess-llm.html
[2]https://www.reddit.com/r/chess/comments/18dr6cp/the_ability_of_gpt_models_at_playing_chess/

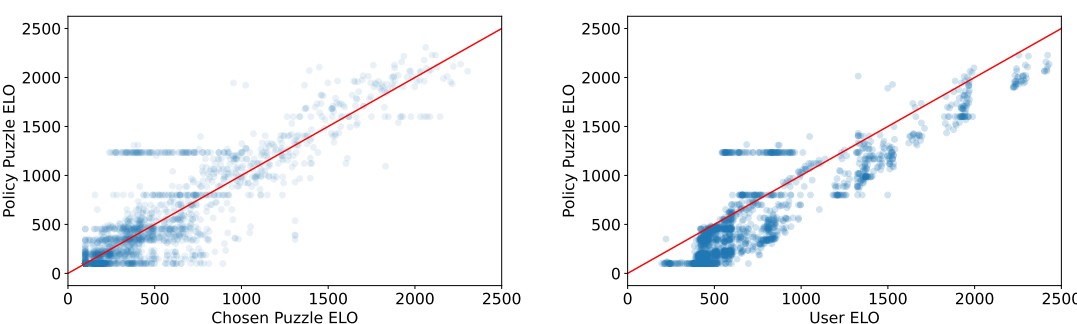

Figure 5: Distribution of trained policy's top recommended puzzle versus the puzzle ELO of the chosen puzzle (left) and the user ELO (right).

## 7.1 POLICY LEARNING

To demonstrate that our policy performs well qualitatively and quantitatively relative to the behavioural policy, we perform evaluations based on importance sampling and visualize and summarize the policy recommendations. As a preliminary qualitative evaluation, we examine the distribution of puzzle ELOs recommended by our policy with respect to the puzzle chosen by chess.com's current policy in Figure 5. As a preliminary sanity check, our policy consistently recommends policies in the rough ballpark of the chosen puzzle and the user ELOs, though our policy tends to recommend harder puzzles on average, especially for users with higher ELO. An interesting trend is that the recommendations typically underestimate the user ELO, which is reflective of the behaviour shown by the behavioural policy as well.

Table 1: Returns obtained by trained policy and behavioural policy through one-step importance sampling.

| ELO | Stagnant Group | | Growth Group | | Average | |
|---|---|---|---|---|---|---|
| | $\pi_\beta$ | $\pi_\theta$ | $\pi_\beta$ | $\pi_\theta$ | $\pi_\beta$ | $\pi_\theta$ |
| 100-600 | $14.3 \pm 1.5$ | $\mathbf{52.9} \pm \mathbf{3.0}$ | $17.7 \pm 0.4$ | $\mathbf{58.8} \pm \mathbf{1.7}$ | $16.9 \pm 0.5$ | $\mathbf{57.4} \pm \mathbf{1.5}$ |
| 600-1000 | $14.1 \pm 1.2$ | $\mathbf{27.0} \pm \mathbf{6.7}$ | $20.9 \pm 0.5$ | $\mathbf{42.3} \pm \mathbf{3.2}$ | $19.5 \pm 0.4$ | $\mathbf{39.2} \pm \mathbf{2.9}$ |
| 1000-1500 | $12.3 \pm 0.4$ | $12.1 \pm 1.2$ | $18.3 \pm 0.6$ | $19.7 \pm 2.4$ | $15.3 \pm 0.4$ | $16.0 \pm 1.4$ |
| 1500+ | $12.2 \pm 0.2$ | $12.2 \pm 1.2$ | $15.9 \pm 0.3$ | $16.1 \pm 1.1$ | $13.5 \pm 0.2$ | $13.6 \pm 0.9$ |

In Table 1, we summarize the results of the one-step importance sampling approaches compared to the behavioural policy for different ELO buckets and growth groups. Across lower ELO buckets (between 100 and 1000), we show statistically significant and consistent improvement compared to the behavioural policy. However, notably, as the ELO increases, our margins of improvement decrease, where our policy is neutral with respect to the behavioural policy.

Qualitatively, this is sensible because optimal puzzle selection is more critical to learning at earlier stages, whereas missteps in puzzle recommendation may not lead to large outcome differences for an experienced chess player. Additionally, an important note is that the vast majority of the user base (and reflected in the training and evaluation data) is concentrated at lower ELO levels, with over 80% of users under 1500 ELO, where our improvement is roughly 110.1%.

While the return is consistently lower for the stagnant group (as is expected), the relative performance improvement from the trained policy is approximately similar across growth groups and stagnant groups. That said, for the smallest ELO bucket, $\pi_\theta$ performs better relatively on stagnant group users, whereas for all others, $\pi_\theta$ performs relatively better on growth group users.

## 7.2 EXPERTS EVALUATION

Using the automated evaluation pipeline that is learned to imitate the judgments of our chess experts, we are able to evaluate the quality of puzzle recommendations made by two models on 200 randomly

selected users' past puzzle-solving history. For the behavoural model $\pi_\beta$, we directly use what was actually recommended to the user at that time. For the trained model $\pi_\theta$, we sample the top 2 puzzles that have the highest probabilities of being recommended.

| Method | Calculation | Pattern recognition |
|---|---|---|
| $\pi_\theta$ (Trained) | 72.39 (8.59) | 61.48 (9.24) |
| $\pi_\beta$ (Behavoural) | 70.17 (14.21) | 58.89 (11.66) |
| $\Delta$ | +2.22 | +2.59 |

Table 2: Comparison of methods for Calculation and Pattern recognition. Values are presented as mean (standard deviation).

| Method | Informativeness | Rating appropriate | Quality | Fun |
|---|---|---|---|---|
| $\pi_\theta$ (Trained) | 75.11 (5.79) | 72.73 (11.58) | 71.31 (5.19) | 70.34 (7.34) |
| $\pi_\beta$ (Behavoural) | 75.11 (7.15) | 67.95 (15.95) | 69.83 (7.29) | 65.57 (8.80) |
| $\Delta$ | 0.00 | +4.77 | +1.48 | +4.77 |

Table 3: Comparison of methods for Informativeness, Rating appropriateness, Quality, and Fun. Values are presented as mean (standard deviation).

For the evaluation, we compute all six metrics. All the metrics have a high variance for the rating of the evaluation pipeline. The difference between the two models is within the margin of error. However, we see signs of positivities that our trained model is able to recognize puzzles that are better than what were served by the default system.

## 8 CONCLUSION AND FUTURE WORK

Our study leverages 1.6 billion puzzle-solving histories to learn the pedagogical value of chess puzzles and develop an automated puzzle selection system using offline reinforcement learning. Validation by titled players supports our model's effectiveness, suggesting a potential for helping many millions of chess players learn chess more efficiently. In the future, we hope to collaborate with our data providers to demonstrate the use of our system and our pipeline has a downstream impact on automating quality teaching material discovery for other domains such as math, language learning, or coding.

## REPRODUCIBILITY STATEMENT

We provide comprehensive details regarding the setup of our experimental setup, ensuring full reproducibility based on the provided information. We are planning to open-source our training code. We provide detailed documentation of the evaluation process, along with all the prompts that we use. We use gpt-4o-2024-08-06 with a max token of 16383 and default temperature. All software used in our evaluation (e.g., DSPy) is open source. However, our data are proprietary owned, although shared with us for the purpose of academic research. Due to such restrictions, we are also not able to open-source our trained model since they can create a potential conflict of business interests for our data provider. However, we plan to apply our model to public dataset of chess puzzles and provide high-quality puzzles identified by our model for free public use.

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
