# OpenReview forum: "Discovering High-Quality Chess Puzzles Through One Billion Plays with Offline Reinforcement Learning"
_ICLR.cc/2025/Conference — ICLR 2025 Conference Withdrawn Submission_

### Official Review · Reviewer_95wq · 2024-10-20

**Soundness:** 2
**Presentation:** 2
**Contribution:** 2
**Rating:** 3
**Confidence:** 4

**Summary:**

The paper presents an offline reinforcement learning system for recommending chess puzzles to facilitate user learning. A causal transformer model is trained on user and puzzle embeddings to predict the next puzzle for the user to play. Experiments show the model is better able to give users puzzles that are appropriate to their skill level than the existing policy implementation. This advantage is particularly large for users at low skill levels, who are the bulk of the population and most sensitive to quitting due to failure.

An additional analysis trains a model to replicate expert annotations of puzzles of various subjective criteria. The model predictions on recommended puzzles show no statistically significant differences when comparing the existing and proposed new model, though results suggest there may be differences in favor of the new model.

**Strengths:**

# originality
Modest.

Recommending content based on policies trained offline is not a technical novelty. There are some details specific to the case that are novel and the chess application is new.

# quality
Modest.

The evaluation of the behavior recommendations shows strong results and is done with a reasonable amount of rigor.

The evaluation by judge ratings is still very preliminary due to limited sample sizes and no clear integration with the rest of the system.

# clarity
Modest.

The paper provides substantial technical detail on the training process. This is strong and facilitates reproduction.

The questions below address specific areas where things were less clear.


# significance
Modest.

The paper will be of interest to applied research in various educational domains (programming, STEM, and so on), and more indirectly of interest to work in offline RL for recommender systems. The ideas can readily be translated to those applications and the scaling efforts would be useful.

**Weaknesses:**

The LLM judging is the biggest weakness for the paper. The results are still preliminary with 200 samples that show no significant differences to the behavior policy (Tables 2 & 3). This makes the content dedicated to this system weakly supported, harming the paper overall. In general the judge proxy model seems better suited to a follow on paper. The space that gets freed up could be devoted to more detailed analyses of the learned policies and more detailed differences in their behaviors. This could also be used to develop a model or metric to predict learning outcomes (ELO changes) based on puzzles a (simulated) user encounters (see below).

# quality
What is the evidence that the reward function will accurately capture ELO changes in real players? The evaluation of the new policy hinges on the assumption this is a direct translation. With any offline study it is important to establish how strong the counter-factual could credibly be. I recognize this is difficult, but without evidence on this point it is hard to gauge how much increases in returns (Table 1) translate into improved learning. Are there canonical metrics in previous work to compare with?

Line 471: What test was used to establish statistical significance? What values were obtained for that test?

# clarity
See below for detailed notes.

**Questions:**

Lines 285-287: Section 4.3 states is it prohibitive to compute the probability normalization across all action embeddings, but in section 4.4 this quantity is computed. Why use this workaround given the embeddings were already computed? Is there some other cost I'm missing?

To what degree does the in-batch negative sample bias the results?
This could be evaluated using a subset of the total data where the true normalization and sample normalization values could be used and the outcomes compared. Knowing how much this method biases the results of the model would be valuable.

Figure 2: Should the yellow and red sets of user and puzzle embeddings both be from $t+1$? I would have expected red to be $t+2$.

Table 1: What is the bold in the table?

Line 481: Does "smallest ELO" here mean "lowest ELO"? This was unclear if it meant "smallest" as in fewest players.
When looking at ELO 100-600 the stagnant group has a mean of 52.9, while growth is 58.8. This does not match the statement that $\pi_{\theta}$ performs better relatively on stagnant group users. Or is relatively meant by the gain relative to $\pi_{\beta}$? That also seems to favor the growth group (gains of 38.6 vs 41.1).

Line 485: Were the 200 users selected across all ELO ratings? How well-balanced was the sample?
This test would benefit from ensuring a balanced sample across buckets. Or only performing it in one or a few buckets separately. Given the importance of low ELO players to the implementation it would be reasonable to examine only the lowest ELO bucket. This might help reduce variance to get more statistically rigorous results.

Tables 2 & 3: These two tables could be combined and pivoted (meaning show the method as columns and the metrics as rows).

Minor typo that spellcheck would miss:
- Line 257: "casual transformer" -> "causal transformer"

---

> ### Author Response · Authors · 2024-12-03
>
> Hi, we are very thankful for your detailed review! This version of the paper does lack a few key ingredients to meet the bar of acceptance. For the future submission, we will improve on what you have suggested:
>
> 1. The human / LLM-as-judge evaluation should have statistically significant results between the two policies. Thank you for proposing several ways to help us achieve that -- which includes looking at users of different Elo buckets.
> 2. Addressing the various minor errors and clarifying the confusion in the text. Thank you for listing them!
>
> Overall, we are grateful for your review, and we will use your suggestions to improve our paper for the next submission.

---

### Official Review · Reviewer_r6oc · 2024-11-03

**Soundness:** 1
**Presentation:** 3
**Contribution:** 2
**Rating:** 3
**Confidence:** 3

**Summary:**

The authors propose a method for automatically generating chess puzzles for learners using a year of user history data and 1.6 billion puzzle-solving histories. They frame this discovery as an offline RL problem, using the ability of a user to solve the puzzle as a reward function and validate a small subset of their data with expert human annotators. They then extrapolate this to the entire pipeline as a whole using the human annotation to direct an LLM.

**Strengths:**

The method the authors used to overcome the exponentially large action space of potential puzzle recommendations is clever and seemingly sound. The question of recommending bite-sized assignments to learners that are appropriate for their mastery of the material is relevant not just to the domain of chess but also to education as a whole.

**Weaknesses:**

Two key major weaknesses of the paper are in the founding assumption and how the effectiveness of the pipeline is evaluated.

1.) The authors propose using whether or not a solver was able to complete the puzzle as a metric for if the puzzle was of an appropriate difficulty for them. However, this feels like an excessive simplification. While it is mitigated by the availability of the difficulty metric (ELO) in respect to the puzzle, it does not seem sound to claim a puzzle was not an appropriate difficulty merely because it was not completed. Steps until completion, time to completion and other fine-grained details may allow for a deeper insight to whether the puzzle was of an appropriate difficulty for the learner. Even this does not guarantee the 'effective long-term increase in knowledge or skill.' mentioned as a heuristic by the authors in lines 61-63.

2.) I am also somewhat skeptical of the ability of LLMs to evaluate the effectiveness of the puzzle, even with the annotation of the human experts to guide them. The results from the training are not in the main paper, and I feel without proof of the effectiveness of the LLMs as judges in this context, the results detailed in the experiments section cannot be adequately judged.

The citing of a reddit thread and blog when there are a number of published works on LLMs and chess feels like the authors did not do a sufficient background study prior to submitting the work.

**Questions:**

1.) How were the categories in Figure 4 picked?

2.) Re 1 in the weaknesses above, can the authors give more justification as to why they chose the specific reward function they did?

3.) From the data, is there a distinct correlation between puzzles done and-or ability to complete puzzles and an increase in a player's ELO?

4.) Re 2 in the weaknesses above, can the authors show the validity of the LLM-judge approach they used? 30 seems too small a number to instill large confidence in the LLMs unless they were able to match the expert annotators numbers exactly, especially given the potential range and variance in chess positions.

4.1.) How much standard deviation was there in the expert annotators' work? Could the authors provide an analysis on the judgement of the annotators and where the points of major agreement or disagreement were?

---

> ### Author Response · Authors · 2024-12-03
>
> Hi, thank you so much for the review and for recognizing this problem as an important problem to solve. We want to quickly address some of the confusion.
>
> > 1.) The authors propose using whether or not a solver was able to complete the puzzle as a metric for if the puzzle was of an appropriate difficulty for them. Steps until completion, time to completion and other fine-grained details may allow for a deeper insight to whether the puzzle was of an appropriate difficulty for the learner.
>
> > Re 1 in the weaknesses above, can the authors give more justification as to why they chose the specific reward function they did?
>
> You made an excellent point here, and we agree. The algorithm we proposed is an RL policy learned through offline data. Ideally, we would like to directly optimize the reward as a player's learning gain (knowledge increase), but that is not accessible (we can only observe if the player solved the puzzle or not, but we don't know how much a player has learned from the puzzle). Therefore, correctness is only a proxy signal that allows us to gauge the quality of the puzzle. Due to data access limitations, we don't have the full player solution history (steps until completion). We are hoping the lack of more fine-grained information can be mitigated by a large amount of player history (>1 billion interactions).
>
> > 2.) I am also somewhat skeptical of the ability of LLMs to evaluate the effectiveness of the puzzle.
>
> We will include a human study to evaluate the puzzles. When we decided to take this approach, it was more along the lines of trying to do a large-scale automated evaluation to understand the difference between policies, letting LLMs imitate human preferences. We will include more details.
>
> Thank you again for the review! The suggestions you made about evaluation will be used to improve our paper in the next iteration (we are withdrawing our paper from the ICLR submission).

---

### Official Review · Reviewer_q4j9 · 2024-11-03

**Soundness:** 3
**Presentation:** 3
**Contribution:** 3
**Rating:** 6
**Confidence:** 3

**Summary:**

The paper considers the design and selection of chess puzzles using a reinforcement learning framework that evaluates puzzles in terms of their contribution to a players' learning and improvement. The paper provides strategies for selecting chess puzzles and evaluates them using techniques including expert annotation.

**Strengths:**

An important aspect of both human and machine interaction with chess as a domain is the design and solving of chess puzzles; they are believed to be an important mechanism for improvement at chess. Given this motivation in terms of improvement, the paper's strategy to explicitly represent this objective within a reinforcement learning framework is very sensible, and it leads to interesting algorithmic insights. The evaluation framework used by the paper provides additional insights into the power of the method.

**Weaknesses:**

The paper would benefit from making a clearer connection between (a) the technical details of the framework being developed and (b) the high-level motivation for selecting puzzles. I would summarize the high-level motivation (b) as roughly saying: certain puzzles will be more valuable than others in helping a player improve; this choice of the "right" puzzle is both player-specific (the best puzzle for one player won't necessarily be the best puzzle for another) and time-specific (the best puzzle for you now won't necessarily be the best puzzle for you later after you've solved some number of puzzles and also (hopefully) improved).

With this distinction in mind:

- It's not clear exactly how these high-level criteria are informing the choice of objective function and algorithm in the technical method itself; the paper would benefit by drawing this connection more explicitly, including more high-level discussion to connect them.

- It's also not clear whether the annotations the experts are asked to provide are the best way to formalize the high-level goals of puzzle selection into a set of annotation instructions. It would be useful ti explain why these particular annotation instructions were chosen.

- Finally, it would help to have some discussion of the causal inference considerations that implicitly underpin the work. In particular, as the authors understand, the fact that the puzzles solved by a rapidly improving chess player are different than the puzzles solved by a stagnating chess player could arise for two very different reasons: (a) because the well-chosen puzzles for the first player are causing them to improve more rapidly, or (b) because a player who naturally has more aptitude or motivation for chess will seek out and solve puzzles more effectively. In (a) the direction of causality works as the paper implicitly suggests -- that by choosing better puzzles we can help them improve -- whereas in (b) the direction of causality works differently -- it simply establishes that people who have more aptitude or motivation for chess will also do better at solving puzzles. Given how central this causal point is to the premise of the paper, there should be more discussion of it, and how it informs the approach.

**Questions:**

Questions:

It would be helpful if the authors could address the weaknesses listed above.

---

> ### Author Response · Authors · 2024-12-02
>
> Hi, thank you for a vote of confidence in our work and the detailed review. We are going to withdraw and rework for a stronger submission.
>
> With that said, we want to address some of the weaknesses.
>
> > It's not clear exactly how these high-level criteria are informing the choice of objective function and algorithm in the technical method itself
>
> The **learning benefit** of each puzzle cannot be easily characterized even by human experts, and the benefit is highly personalized (a puzzle valuable to one player might not be to another player, or at one time it's valuable but less so at a different time). The criteria are designed as a post-evaluation to understand whether the learned policy is useful or not. But you are completely right in the sense that this can be confusing -- if we have an objective measure of the quality of the puzzle, why isn't that the reward for the RL policy? We will spend more time to think about this.
>
> > It's also not clear whether the annotations the experts are asked to provide are the best way to formalize the high-level goals of puzzle selection into a set of annotation instructions. It would be useful to explain why these particular annotation instructions were chosen.
>
> We went through a few rounds of designs. Will include more details on why they are chosen.
>
> > Finally, it would help to have some discussion of the causal inference considerations that implicitly underpin the work.
>
> >  (a) because the well-chosen puzzles for the first player are causing them to improve more rapidly, or (b) because a player who naturally has more aptitude or motivation for chess will seek out and solve puzzles more effectively.
>
> The dataset we curated is neither constructed in (a) nor in (b). The platform (chess.com) is randomly serving puzzles to players within a given elo bucket. The player cannot seek out or choose puzzles (at least not in our dataset). Chess.com also doesn't "choose" intentionally to make the player improve. Our dataset is made of different random permutations of puzzle sequences. Our goal is to identify which sequence is more beneficial to the player.
>
> But we agree that it would be much better if we could make this point extremely clear.
>
> Thank you again for the review! Even though we are going to withdraw our paper, we are still happy to discuss it with you through the comment system. We are hoping to make a strong submission in the near future. Thank you for your help in improving our work!

---

### Official Review · Reviewer_hnZs · 2024-11-07

**Soundness:** 2
**Presentation:** 1
**Contribution:** 2
**Rating:** 3
**Confidence:** 4

**Summary:**

The authors look to use RL to recommend chess puzzles from a fixed set, with the RL objective being that the player gets the puzzle correct and that it has a high Elo. They also add some human annotations and do something with LLMs to reproduce them.

**Strengths:**

The RL algorithm implemented in the paper is novel (at least in the chess space) and the puzzle evaluations by experts is a new dataset.

# originality

Moderate, the RL approach appears to be new, but the underlying optimization is the same as what chess.com is already doing with their expanding Elo bands (described in line 098). The short and incomplete related works section means I may be overestimating the originality though.

# quality

Moderate, the writing is mostly OK, but there are a few weird or difficult to follow sections. I'm also not clear what the LLM section contributes to this.

# clarity

This is similarly limited by the writing style. The results were difficult to understand as there is no real baseline to compare to, so I don't know if the numbers (and thus the main results) are meaningful. I suggest using the chess.com "algorithm" as a baseline.

I also found the scatter plots difficult to read as they look to be trying to show density, where an actual density plot would be preferable.

# significance

low, the RL objective function is the key to the entire paper is not new so while the RL design might be it's not a major result.

**Weaknesses:**

I find this paper unconvincing, they don't compare to a reasonable baseline or previous works so evaluating the numbers directly is impossible, the LLMs section seems tacked on, and they don't run user studies despite claiming to "automatically choose a set of puzzles to better support chess learners".

The value function they propose also doesn't seem to optimize for what they are looking for (learning), it just optimizes for success on higher Elo puzzles. If the authors could explain how this leads to learning that would greatly improve my valuation of the paper.

I'm also concerned by the lack of code release. The authors promise they will release the code but not the models or data. What part of the data are "proprietary owned"? There are other papers published on the chess.com puzzles dataset[1] so I'm not clear what is the limiting factor.


[1] Anderson, Ashton, Jon Kleinberg, and Sendhil Mullainathan. "Assessing human error against a benchmark of perfection." ACM Transactions on Knowledge Discovery from Data (TKDD) 11.4 (2017): 1-25.

**Questions:**

+ Figure 1b is not very clear, consider a density plot, giving a regression, or even using an alpha lower than 1 for the points.
+ ELO is not a thing, I believe you mean Elo, although which one is never specified
+ There is previous work out of ICLR last year looking at a related problem[1]
+ "Anecdotally, LLMs seem .." there are research papers on LLMs to play chess [2] why was reddit used instead?

[1] Hamade, K., McIlroy-Young, R., Sen, S., Kleinberg, J. and Anderson, A., Designing Skill-Compatible AI: Methodologies and Frameworks in Chess. In The Twelfth International Conference on Learning Representations.
[2] Feng, Xidong, et al. "Chessgpt: Bridging policy learning and language modeling." Advances in Neural Information Processing Systems 36 (2024).

---

> ### Author Response · Authors · 2024-12-02
>
> Hi, we are very thankful for your detailed review! This version of the paper does lack a few key ingredients to meet the bar of acceptance. For the future submission, we will improve on what you have suggested:
>
> 1. Adding a user study to validate the model
> 2. Integrate the RL and LLM better
> 3. Compare to the original chess.com baseline
> 4. Explore whether it's possible to release data (thanks for mentioning past work have already released data from chess.com -- we will definitely talk to our data provider)
>
> Do you find the RL algorithm optimizing for elo increase unconvincing for learning gains? Elo increase is a proxy measure for a player's learning progress, similar to optimizing for student test scores.
>
> Regardless, we appreciate the suggested relevant literature. Our work was heavily inspired by McIlroy-Young's earlier work and we hope to improve what we have and share with everyone in the future.

---

### Note · Authors · 2024-12-04

**Comment:**

We thank all the reviewers for their review and decided to incorporate their feedback into the next round of submission. The amount of feedback we received is incredible, and we sincerely thank the AC and all reviewers for their work!

**Withdrawal Confirmation:**

I have read and agree with the venue's withdrawal policy on behalf of myself and my co-authors.